# Peer review of "HIV Disclosure to Infected Children Involving Peers: A New Take on HIV Disclosure in the Democratic Republic of Congo"

_children, 2023, doi:10.3390/children10071092_

Round 1

Reviewer 1 Report

The manuscript consists of total 20 pages, including 3 tables and the list of total 38 literature references. The article presents the original results of the study on how the information on HIV infection is revealed to children. As HIV infection is one of major problems in African countries, it is both current and important and as such also likely to raise the interest of the Journal's Readers. However, the title of the article seems unclear to me and shall be modified into e.c. "New approach to disclosure of information on infection to HIV-positive children involving peers evaluated in the Democratic Republic of Congo". The manuscript has a logical structure, the line of argumentation is coherent and English language quality of the text is acceptable.

The Abstract mirrors the key information provided in the main text of the manuscript adequately.

The Introduction provides enough information on the context of the problem to justify why the study was undertaken.
The Material and methods are described clearly and in enough detail.
The Results are consistent with the declared methodology, clearly presented and adequately supported by data in tables.
The Discussion presents the interpretation of the presented results and places them in the broader knowledge context on the topic.
The separate Conclusion section is absent, which shall be added.
The tables are clearly constructed and adequately captioned.
The literature references are relevant to the topic of the article and recent enough. However, the Authors shall consider enriching their text by rising the following aspects in the introduction or discussion:
- developmental aspects in youth living with HIV as in e.c. https://doi.org/10.3390/children10050798 , https://doi.org/10.3390/children10020405 , https://doi.org/10.3390/ijerph20042996 , https://doi.org/10.3390/v13101947 , https://doi.org/10.3390/children7120289
- psychological aspects of presence of HIV infection risk awareness in children as in e.c. https://doi.org/10.3390/ijerph20032499
- disclosure of HIV infection to children as in e.c. https://doi.org/10.3390/children9121989 , https://doi.org/10.3390/children9121955 , https://doi.org/10.3390/children9081239 , https://doi.org/10.3390/medicina55080433 , https://doi.org/10.3390/ijerph16040595
- perspective of parents of HIV-positive children as e.c. in https://doi.org/10.3390/ijerph19116879 , https://doi.org/10.3390/vaccines9111331 , https://doi.org/10.3390/ijerph16173162

The manuscript has a logical structure, the line of argumentation is coherent and English language quality of the text is acceptable, though still some style improvements may be considered.

Reviewer 2 Report

HIV disclosure to infected children involving peers: a new take on HIV disclosure in the Democratic Republic of Congo

This is an important contribution about appropriately informing HIV-infected children of their in sub-Saharan Africa. Still, the article requires several significant changes before it can be considered fitted for publication:

1.     Abstract: adhere to a structured format, which includes background, objectives, materials and methods, results, and conclusion.

2.     Line 66: disclosure rates of HIV-status disclosure to children/adolescents living with HIV in sub-Saharan Africa (SSA) are generally low, ranging from 9 to 72%, and constitute a significant public health challenge [3]. Why? Please explain.

3.     What specific circumstances of the cultural context of the DR Congo exist in relation to this matter?

4.     Objectives of the study must be clearly stated.

5.     How are cognitive skills and emotional maturity of the patients assessed? Please explain.

6.     Please rewrite the article to include a materials and methods section, and procedures, before presenting the results.

7.     Table 2. References?

8.     Reliability information regarding the measures used must be communicated.

9.     Authors should conduct more robust statistical analyses to deepen the results. I suggest correlational matrixes and logistic regressions.

10.  Include specific circumstances of the cultural context of the DR Congo when discussing the results.

11.  Please discuss epidemiological and/or health implications of these results.

Best wishes.

Round 2

Reviewer 2 Report

Thank you for implementing all the requested changes, it has very much improved the overall quality of the article.

Best wishes.